# Wafer-Bonding Fabricated CMUT Device with Parylene Coating

**DOI:** 10.3390/mi12050516

**Published:** 2021-05-04

**Authors:** Changde He, Binzhen Zhang, Chenyang Xue, Wendong Zhang, Shengdong Zhang

**Affiliations:** 1School of Electronic and Computer Engineering, Peking University, Shenzhen 518055, China; hechangde@nuc.edu.cn; 2Institute of Microelectronics, Peking University, Beijing 100871, China; 3State Key Laboratory of Dynamic Testing Technology, North University of China, Taiyuan 030051, China; zhangbinzhen@nuc.edu.cn (B.Z.); xuechenyang@nuc.edu.cn (C.X.); wdzhang@nuc.edu.cn (W.Z.)

**Keywords:** capacitive micromachined ultrasound transducer (CMUT), underwater imaging, fusion bonding, pulse-echo signal, micromachine, transducer, Parylene

## Abstract

The advantages of the capacitive micromachined ultrasound transducer (CMUT) technology have provided revolutionary advances in ultrasound imaging. Extensive research on CMUT devices for high-frequency medical imaging applications has been conducted because of strong demands and fabrication realization by using standard silicon IC fabrication technology. However, CMUT devices for low-frequency underwater imaging applications have been rarely researched because it is difficult to fabricate thick membrane structures through depositing processes using standard IC fabrication technology due to stress-related problems. To address this shortcoming, in this paper, a CMUT device with a 2.83-μm thick silicon membrane is proposed and fabricated. The CMUT device is fabricated using silicon fusion wafer-bonding technology. A 5-μm thick Parylene-C is conformally deposited on the device for immersion measurement. The results show that the fabricated CMUT can transmit an ultrasound wave, receive an ultrasound wave, and have pulse-echo measurement capability. The ability of the device to emit and receive ultrasonic waves increases with the bias voltage but does not depend on the voltage polarity. The results demonstrate the viability of the fabricated CMUT in low-frequency applications from the perspectives of the device structure, fabrication, and characterization. This study presents the potential of the CMUT for underwater ultrasound imaging applications.

## 1. Introduction

Ultrasound imaging technology has a wide range of applications, including medical diagnostics, underwater exploration, and nondestructive evaluation of materials. Ultrasound imaging is based on the scattering of ultrasound energy by material interfaces with different properties through interactions governed by acoustic physics. Ultrasound transducers are used to transmit and receive acoustic energy. Since the first demonstration of a capacitive micromachined ultrasound transducer (CMUT) in the early 1990s [1,2,3], the CMUT has become a promising candidate for three-dimensional (3D) ultrasound imaging systems [4,5,6]. The most attractive advantages of CMUTs over traditional piezoelectric ultrasonic transducers are a wide bandwidth in immersion and an easy realization of two-dimensional (2D) high-density arrays for real-time ultrasonic volumetric imaging [4,7,8,9,10,11,12]. The CMUTs also provide advantages of the reduction in size and cost and integration with electronics [13,14].

In the past 3 decades, extensive research has been conducted on modeling [15,16], performance improvement [17,18,19], fabrication methods [9,20,21,22,23,24], front-end circuitry [4,8,10,25,26,27,28], and applications [4,5,11,12,29,30,31,32] of the CMUT technology. The pioneering research on the CMUT mostly focused on medical ultrasound imaging applications and less consideration was given to underwater ultrasound imaging applications. The main differences between these two types of applications relate to different requirements for device frequencies. The device’s structural parameters, materials, and fabrication process can also differ. The CMUT devices for medical imaging are usually designed to operate at frequencies over 3 MHz and are fabricated by the surface micromachined process. Silicon nitride (Si_3_N_4_) film deposited by Plasma Enhanced Chemical Vapor Deposition (PECVD) or (Low Pressure Chemical Vapor Deposition (LPCVD) is a vibration membrane material of the device, and its thickness is less than 1.5-μm in medical CMUT. The size of a device for underwater imaging is much larger than that of the medical CMUT. The CMUT devices for underwater imaging are usually designed to operate at frequencies below 1.5 MHz, and the membrane thickness is wider than 2-μm. However, with the increase of thickness, it is more difficult to fabricate a low stress silicon nitride membrane by standard silicon integrated circuit (IC) fabrication technology, therefore mono-crystalline silicon or polysilicon is a good choice as a vibration membrane material. Although traditional piezoelectric transducers can meet the requirements of linear arrays due to the relatively large size of transducers for underwater acoustic imaging, it is difficult to fabricate 2D transducer arrays. Fabrication of 2D transducer arrays is an inherent advantage of the CMUT technology, which has been demonstrated in high-frequency medical imaging applications [4,5,12]. Additionally, mass-produced CMUT transducers can reduce the cost of devices. In addition, high uniformity of array elements based on the silicon micromachined process can reduce the complexity of back-end data processing, while the broadband characteristics of devices can improve the imaging quality of the system. Due to the mentioned advantages of the CMUT technology, the CMUT devices are highly expected to be applied to underwater imaging systems.

This article aims at the potential of the CMUT devices for underwater ultrasound imaging applications. A CMUT device with a 2.83-μm thick silicon membrane is proposed and fabricated. The CMUT device is fabricated using silicon fusion wafer-bonding technology. The mono-crystalline silicon is selected as a vibration membrane material because of its stable, certain mechanical properties. The pulse-echo underwater measurement of the device is conducted to demonstrate the viability of the fabricated CMUT device.

## 2. Materials and Methods

### 2.1. Transducer Structure

A CMUT element usually comprises several capacitor cells connected in parallel. Each cell is a parallel plate capacitor with a movable membrane over a vacuum sealed cavity. The frequency of the transducer affects the imaging resolution and the detection range. The operation frequencies between 300 kHz and 2.5 MHz are mostly needed for different underwater imaging applications. The structure properties of the movable membrane over vacuum cavity determine the frequency of the CMUT element. The movable membrane is designed to be a circular membrane, which can be equivalent to a circular thin plate. The lowest resonant frequency can be expressed as:(1)$$f_{0}=\frac{0.47t_{m}}{a^{2}}\sqrt{\frac{E}{\rho \left(1-\sigma^{2}\right)}},$$
where *E* is Young’s modulus, *ρ* is the density of the membrane material, *σ* is the Poisson’s ratio, a is the radius of the circular membrane, and *t_m_* is the thickness of the membrane. A CMUT transducer for low-frequency underwater imaging applications is designed. The parameters of the CMUT transducer are given in Table 1. The structure of the cell is shown in Figure 1. The cell consists of two conductive silicon plates separated by the vacuum gap dielectric and oxide silicon dielectric. One plate is a thin metalized silicon membrane used as a top electrode of a capacitor, and the other plate is a heavily doped silicon substrate used as a bottom electrode. A thickness of 2.83-μm and a diameter of 180-μm of the silicon membrane are the optimized design parameters to get 1.4 MHz center frequency. The theoretical center frequency is 1.43 MHz, calculated from Equation (1). Vacuum cavity gap distance is another critical parameter and 0.65-μm height is designed to balance the transmitting and receiving performance of the CMUT. The oxide silicon insulation layer of 0.15-μm thickness between the vacuum cavity and substrate is used to prevent the silicon membrane from contacting and short-circuiting with the silicon substrate when the silicon membrane collapses under high voltage. Another oxide silicon insulation layer on the silicon membrane is also designed to prevent short-circuiting and improve withstanding voltage performance. The CMUT transducer comprises 900 capacitor cells connected in parallel on the same die, as shown in Figure 2, where the 3D measuring laser microscope image of the fabricated CMUT device is displayed. 

### 2.2. Capacitive Micromachined Ultrasound Transducer (CMUT) Fabrication

The CMUT device was fabricated using silicon fusion wafer-bonding technology, which was a direct bond between two silicon surfaces at high temperatures and showed features of high bond strength and extreme hermeticity [33]. The wafer-bonding was used because it allows easier fabrication of membrane structures from mono-crystalline silicon. Mono-crystalline silicon as a mechanical material has been extensively studied and is well characterized. The process began with two wafers: a prime quality oxide mono-crystalline silicon wafer and a Silicon-On-Insulator (SOI) wafer. The wafer-bonding fabrication process was described in detail in earlier reports [21,22], therefore, in this article, the fabrication process of the designed CMUT will be summarized and introduced briefly.

First, the oxide layer was etched 650 nm away from the total 800-nm thickness to define cavities on the oxide mono-crystalline silicon wafer, as shown in Figure 3b. Next, the SOI wafer and the cavities patterned oxide silicon wafer were pre-bonded in the vacuum at room temperature, and the pre-bonded wafers were annealed at high temperature, with a maximum temperature up to 1150 °C, which denoted the fusion bonding process, as shown in Figure 3c. The handle layer and the buried oxide (BOX) layer of the SOI wafer were then removed to release the thin membranes of the device layer by performing the Chemical Mechanical Polish (CMP) step and wet etching, as shown in Figure 3d. At that point, an SOI wafer with cavities was fabricated and a 100-nm thick oxide silicon layer was deposited on the device layer, as shown in Figure 3e. After that, around the device cells, the patterned silicon membrane trenches were defined and etched to the oxide layer to isolate the device electrically from the other devices and frame structure. Finally, the aluminum metal layer was sputtered on the side of the device layer of the cavity SOI and patterned, thus forming the top electrode of the CMUT device. Another aluminum metal layer was sputtered on the side of the handle layer of the cavity SOI but not patterned, thus forming the bottom electrode of the CMUT device, as shown in Figure 3f. Annealing was performed to make ohmic contact between the bottom electrode and highly-doped silicon substrate.

### 2.3. CMUT Package for Immersion Measurement

The fabricated CMUT die was glued to a printed circuit board (PCB). The whole bottom electrode of the die was glued to the pad on the PCB by conductive epoxy, and the pad of the top electrode was wire-bonded to another pad on the PCB. In order to make CMUT work underwater, a waterproof package was needed. Parylene-C is a protective polymer material and is widely used for waterproofing sensors. After gluing and wire-bonding the die, a 5-μm thick Parylene-C was conformally deposited on the PCB by vapor deposition in order to seal the device, PCB, and wires for immersion operation. The CMUT device with the Parylene-C coating is shown in Figure 4.

## 3. Results

### 3.1. Capacitance-Voltage Characteristics

The CMUT’s operating principle is based on the elastic deformation of the thin plates of the capacitors under electrostatic force or acoustic wave. The thin plate was composed of the top electrode, oxide silicon insulating dielectric layer, and the silicon membrane. The capacitance-voltage (CV) characteristics of the CMUT device directly illustrate the quantitative relationship between the structure deformation and bias voltage. The CV curve is also related to the device’s important parameters, such as sensitivity, electromechanical conversion efficiency, and collapse voltage. The CV measurements were performed using the KEYSIGHT E4990A impedance analyzer, and bias voltage swept from −40 V to 40 V. The CV curves are presented in Figure 5. The black curve was obtained from the measurement results of the CMUT without Parylene-C on the PCB, and the red curve was obtained from the results of the same CMUT with deposited 5-μm thick Parylene-C.

The results showed that depositing the Parylene affected the CV characteristics of the CMUT device. The capacitance of the CMUT without Parylene increased by 460 pF when the bias voltage changed from 0 to 40 V, while the capacitance of the CMUT with Parylene increased by 402 pF under the same conditions. There was a slight distortion at +10 V in the curves, and this phenomenon did not occur at −10 V. The silicon membrane and silicon substrate of the CMUT capacitor were both semiconductor materials. The parasitic capacitance, due to the semiconductor material, caused this distortion and inconsistency.

### 3.2. Transmitting Performance

In the measurement of the CMUT transmitting performance, the CMUT was driven by a power amplifier (HSA4101, NF Corporation, Kanagawa, Japan) and emitted ultrasound waves in the water tank. The ultrasound output generated by the CMUT was measured with a calibrated needle hydrophone (NH4000, Precision Acoustics Ltd, Dorchester, UK). The distance between the CMUT and hydrophone was 2.7 cm. The measurement setup of the CMUT transmitting performance is shown in Figure 6. The needle hydrophone was fixed on the three-dimensional displacement platform. By finely adjusting the horizontal position, vertical position, and angle of the hydrophone, the maximum output voltage amplitude was found and the output response signal of the hydrophone at this position was recorded.

Figure 7a shows the output responses of the needle hydrophone when the CMUT was driven by the sinusoidal impulses voltage signals with the DC bias voltages of −15 V, −20 V, and −25 V. Sinusoidal impulses consisted of five sine cycles with a fixed peak-to-peak voltage amplitude of 20 V and fixed frequency of 1.1 MHz. Figure 7b shows the output responses of the needle hydrophone when the CMUT was driven by the sinusoidal impulse voltage signals, with the DC bias voltages of 15 V, 20 V, and 25 V. The response peak-to-peak output amplitude of the needle hydrophone was 59.1 mV at the bias voltage of −15 V, 75.1 mV at the bias voltage of −20 V, 94.2 mV at the bias voltage of −25 V, 60.1 mV at the bias voltage of 15 V, 75.1 mV at the bias voltage of 20 V, and 95.2 mV at the bias voltage of 25 V. Figure 7c shows that the output amplitudes of the hydrophone at negative and positive bias voltages were basically the same at the same absolute voltage. The output responses indicated that the transmitting performance of the designed CMUT was related to the absolute DC bias voltage amplitude but not the voltage polarity, which is consistent with the fact that the change of capacitance is related to absolute voltage and not polarity. Additionally, the output amplitudes of the needle hydrophone reflect the transmitting sensitivity of the CMUT transducer. In the non-collapse working mode, transmitting sensitivity of CMUT increased with the increase of the bias voltage, as shown in Figure 7c, which is consistent with the theory. 5-cycle pulse signals were clearly visible in the output response of the needle hydrophone, as shown in Figure 7a,b, which proves that the CMUT dynamically responsed perfectly to the 5-cycle sine excitation signals at different bias voltages.

### 3.3. Receiving Performance

The CMUT receiving performance was evaluated at various DC bias voltages using a constant-transmitting acoustic source. The transmitting acoustic source was another CMUT device (CMUT-2#) that had the same structure as the CMUT to be evaluated (CMUT-1#). The distance between the CMUT-1# and the CMUT-2# was 9.6 cm. The CMUT-2# was driven by the power amplifier with five sinusoidal impulses signals, with a peak-to-peak amplitude of 20 V and a frequency of 1.1 MHz added to the DC bias voltage of 20 V. The DC bias voltage was applied to the CMUT-1# by a bias tee circuit. The current output of the CMUT-1# device flowed through the bias tee and was amplified and converted to the voltage output by a trans-impedance amplifier (TIA) circuit. The TIA circuit consisted of a broadband low-noise amplifier chip (AD, LTC6268-10). The voltage output of the TIA circuit was the output response amplitude of the CMUT-1# device. The measurement setup of the CMUT receiving performance is shown in Figure 8, and the measurement results are shown in Figure 9. The peak-to-peak output response amplitude of the CMUT-1# was 1559.4 mV at the bias voltage of −15 V, 1902.2 mV at the bias voltage of −20 V, 2364.1 mV at the bias voltage of −25 V, 1523.7 mV at the bias voltage of 15 V, 1935.5 mV at the bias voltage of 20 V, and 2467.9 mV at the bias voltage of 25 V.

As shown in Figure 9c, the output amplitude of the CMUT-1# increased with the absolute voltage value. The difference of the output amplitudes of CMUT-1# at negative and positive bias voltages was small at the same absolute voltage. The maximum output voltage difference was only 103.8 mV, and the voltage difference when the bias voltages of −25 V and 25 V were applied was only 4.39%. The output responses indicate that the receiving performance of the designed CMUT is related to the absolute DC bias voltage amplitude but not the voltage polarity. It was demonstrated again that the change of capacitance was related to the absolute voltage and irrelevant to the polarity. 5-cycle pulse signals were clearly visible in the output response of the CMUT-1#, as shown in Figure 8a,b, which proves that the CMUT-1# dynamically responsed to the ultrasound waves at different bias voltages. In the non-collapse working mode, the output amplitudes of CMUT-1# increased with the increase of the bias voltage, as shown in Figure 9c, which is consistent with the theory.

### 3.4. Pulse-Echo Measurements

The measurement setup of the CMUT pulse-echo signal is shown in Figure 10. The CMUT device emitted ultrasound signals and received the ultrasound echo signals reflected from the aluminum block. The distance between the CMUT and the aluminum block was 5.1 cm. The high voltage bipolar pulses were generated by the pulse generator circuit MAX14808. The two echo signals reflected by the aluminum block are shown in Figure 11a. Figure 11b shows the first echo signals when the CMUT device was biased at 15 V, 20 V, and 25 V and excited by three bipolar pulses (9.4 V, 1.1 MHz) at a pulse repetition frequency of 500 Hz. Figure 11c shows that the first echo signal amplitude increased with the DC bias voltage applied to the CMUT device. The pulse signals were clearly visible in the output response of the CMUT, as shown in Figure 9b, which proves that the CMUT dynamically transmitted ultrasound excited by pulse excitation signals and received the ultrasound waves reflected from the aluminum block at different bias voltages. The output amplitudes of the CMUT increased with the increase of the bias voltage, as shown in Figure 9c, which is consistent with the theory.

## 4. Discussion

This paper studies the potential of CMUT devices for underwater ultrasound imaging applications and demonstrates the viability of the proposed CMUT device from the perspectives of the device structure, fabrication, and characterization. A CMUT device operating at a frequency below 1.5 MHz that meets the demands of underwater imaging applications is proposed and fabricated. The transmitting, receiving, and pulse-echo characteristics of the fabricated CMUT device were measured at the frequency of 1.1 MHz in the water tank. The results show that the 5-μm thickness Parylene-C coating on the CMUT device meets the waterproof requirements of underwater measurement.

The designed CMUT can transmit and receive ultrasound waves separately. The designed CMUT can also receive the echo signal of the ultrasonic wave emitted by itself when the emitting ultrasonic wave is reflected from obstacles. The output response of the proposed CMUT is related to the absolute bias voltage but not the bias voltage polarity. The ability of the proposed device to emit and receive ultrasonic waves increases with the bias voltage, but the applied bias voltage needs to be within the device tolerance range. The optimal bias voltage and the maximum bias voltage of the CMUT device are not studied in this work, but these two parameters are important and will be analyzed in our future research.

In the fabrication process of the proposed CMUT device, the silicon fusion wafer-bonding process is the key technology. The silicon fusion wafer-bonding process allows easier fabrication of thick membrane structures, which are required by low-frequency CMUT devices. The widely studied surface micromachined process is used to fabricate the CMUT devices by deposition, and it is suitable for the fabrication of high-frequency CMUT devices. However, it is difficult to deposit a thick membrane structure due to stress-related problems, but a thick membrane structure is required for low-frequency CMUT devices.

It should be noted that the CMUT technology is not simply a low-cost replacement of the piezoelectric transducer technology. Many features of the CMUT technology enable revolutionary advances in ultrasound imaging. Recently, extensive research on the CMUT has focused on its application in medical imaging systems. There are a few studies on CMUTs for underwater imaging applications. The realization of 3D underwater imaging sonar depends on 2D transducer arrays. The CMUT technology is the most promising technology to fabricate 2D transducer arrays. Based on the results presented in this article, the CMUT devices could play an important role in future 3D imaging sonar systems.

## 5. Conclusions

A CMUT device with a 2.83-μm thick silicon membrane is proposed and fabricated using silicon fusion wafer-bonding technology. A 5-μm thick Parylene-C is conformally deposited on the device for immersion measurement. The transmitting and receiving performance of the device were evaluated at various DC bias voltages. The results presented in this article demonstrate that the fabricated CMUT device with Parylene coating meets the requirements of underwater measurement from the perspective of structure, fabrication, and characterization. Based on the device’s structural parameters and fabrication process proposed in this article, the CMUT transducer arrays can be developed and used in underwater imaging sonar systems.

## Figures and Tables

**Figure 1 micromachines-12-00516-f001:**
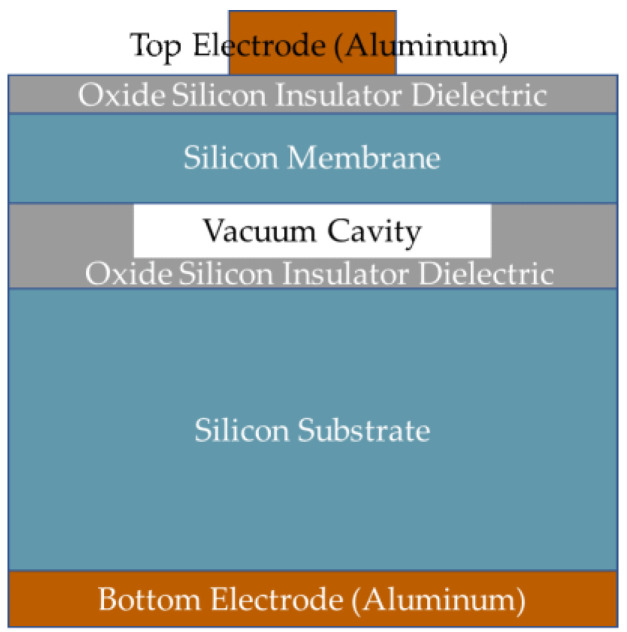
Cross-section of a single capacitive micromachined ultrasound transducer (CMUT) cell.

**Figure 2 micromachines-12-00516-f002:**
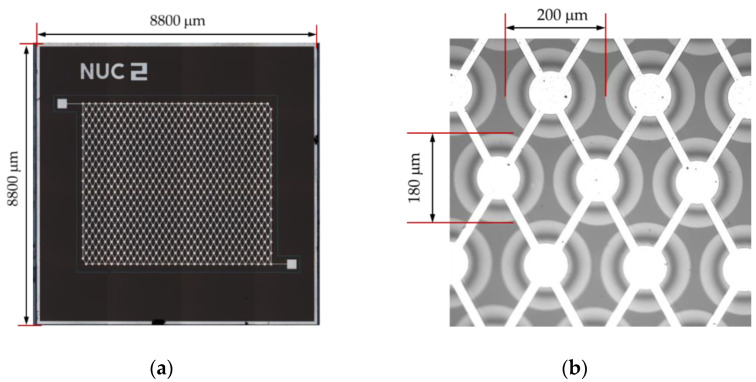
A microscope photograph of the fabricated CMUT device die. (**a**) Top view of the fabricated CMUT device with 900 cells; (**b**) magnified view of cells.

**Figure 3 micromachines-12-00516-f003:**
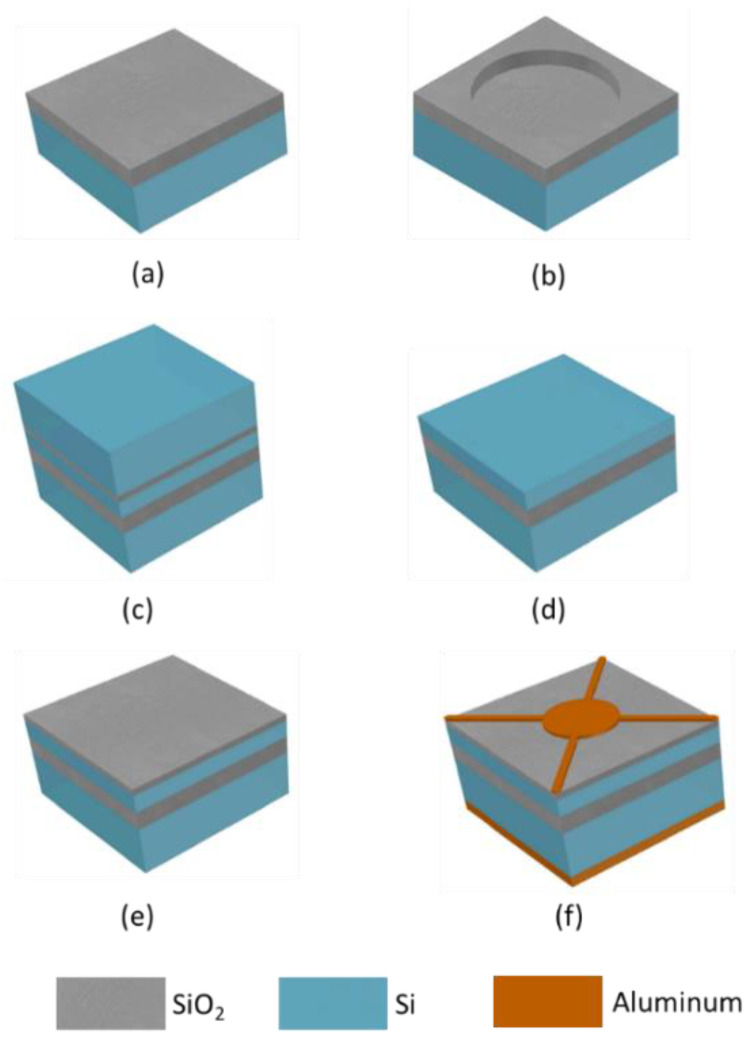
The CMUT fabricated using the silicon fusion wafer-bonding technology. (**a**) Oxide mono-crystalline silicon wafer; (**b**) cavity definition; (**c**) wafer-bonding of the oxide mono-crystalline silicon wafer and Silicon-On-Insulator (SOI) wafer; (**d**) removal of the handle and the buried oxide (BOX) layer of the SOI wafer to release the membranes; (**e**) depositing a 100-nm thick oxide silicon layer as an insulator dielectric; (**f**) fabrication of the top and bottom electrodes.

**Figure 4 micromachines-12-00516-f004:**
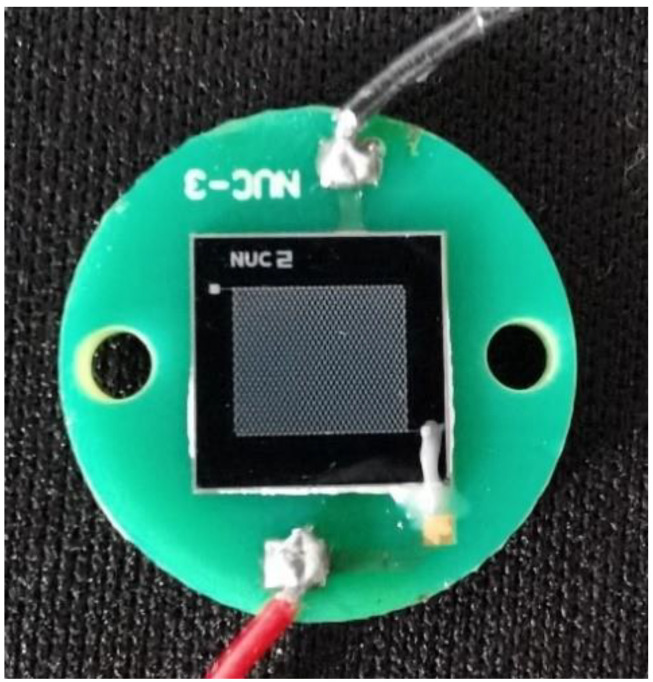
Photograph of the CMUT glued and wire-bonded to the printed circuit board (PCB).

**Figure 5 micromachines-12-00516-f005:**
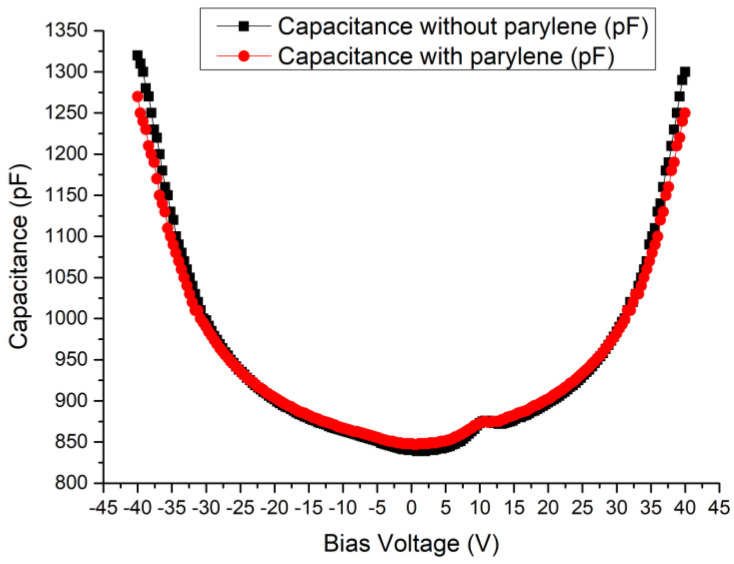
The capacitance-voltage (CV) curves of the CMUT device.

**Figure 6 micromachines-12-00516-f006:**
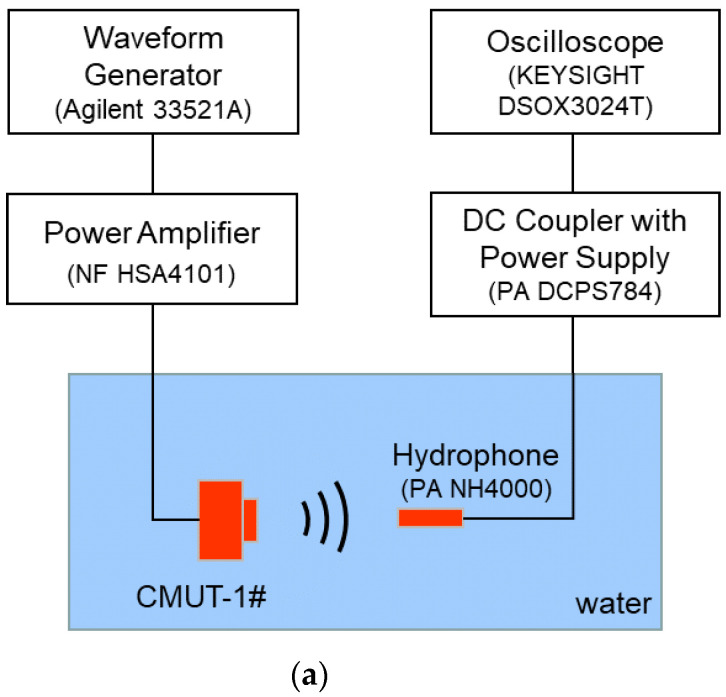
(**a**) Schematic of the measurement setup of the CMUT transmitting performance; (**b**) photograph of the measurement setup.

**Figure 7 micromachines-12-00516-f007:**
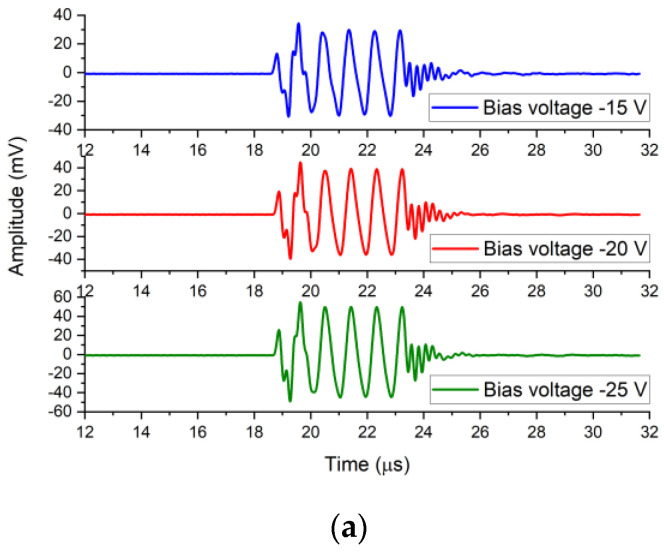
(**a**) Response waveforms of the hydrophone when the CMUT was driven by the DC bias voltages from −15 V to −25 V; (**b**) response waveforms of the hydrophone when the CMUT was driven by the DC bias voltages from 15 V to 25 V; (**c**) peak-to-peak response amplitude of the hydrophone when the CMUT was driven by different DC bias voltages.

**Figure 8 micromachines-12-00516-f008:**
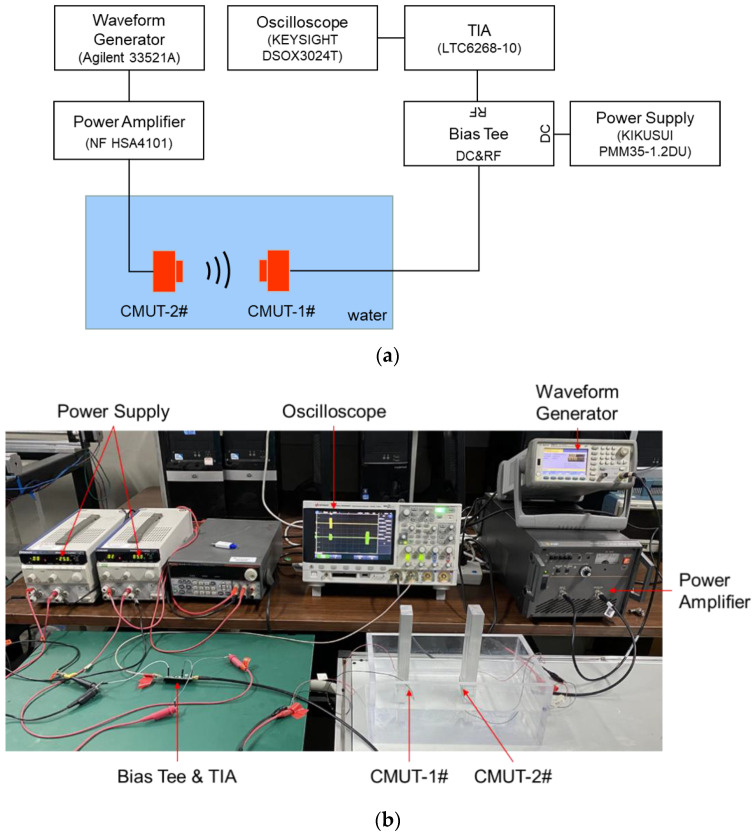
(**a**) Schematic of the measurement setup of the CMUT receiving performance; (**b**) photograph of the measurement setup.

**Figure 9 micromachines-12-00516-f009:**
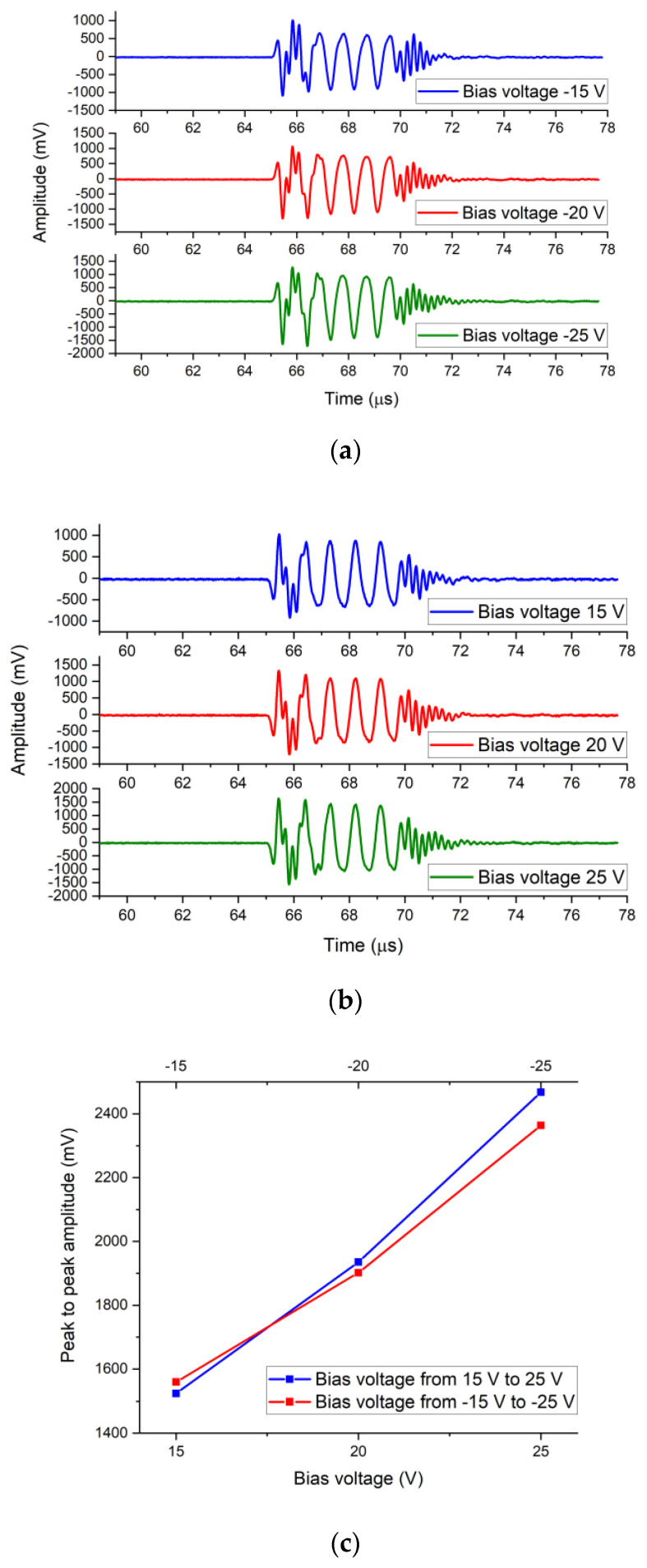
(**a**) Response waveforms of the CMUT-1# driven by the DC bias voltages from −15 V to −25 V; (**b**) response waveforms of the CMUT-1# driven by the DC bias voltages from 15 V to 25 V; (**c**) peak-to-peak response amplitude of the CMUT-1# driven by different DC bias voltages.

**Figure 10 micromachines-12-00516-f010:**
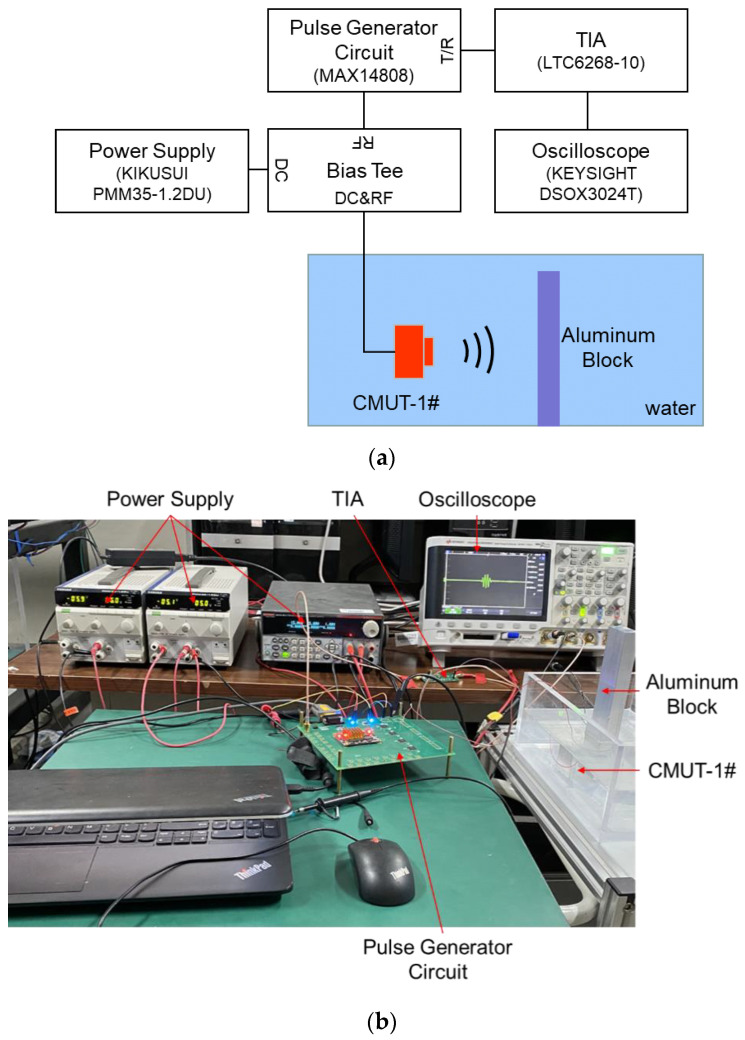
(**a**) Schematic of the measurement setup of the CMUT pulse-echo signal; (**b**) photograph of the measurement setup.

**Figure 11 micromachines-12-00516-f011:**
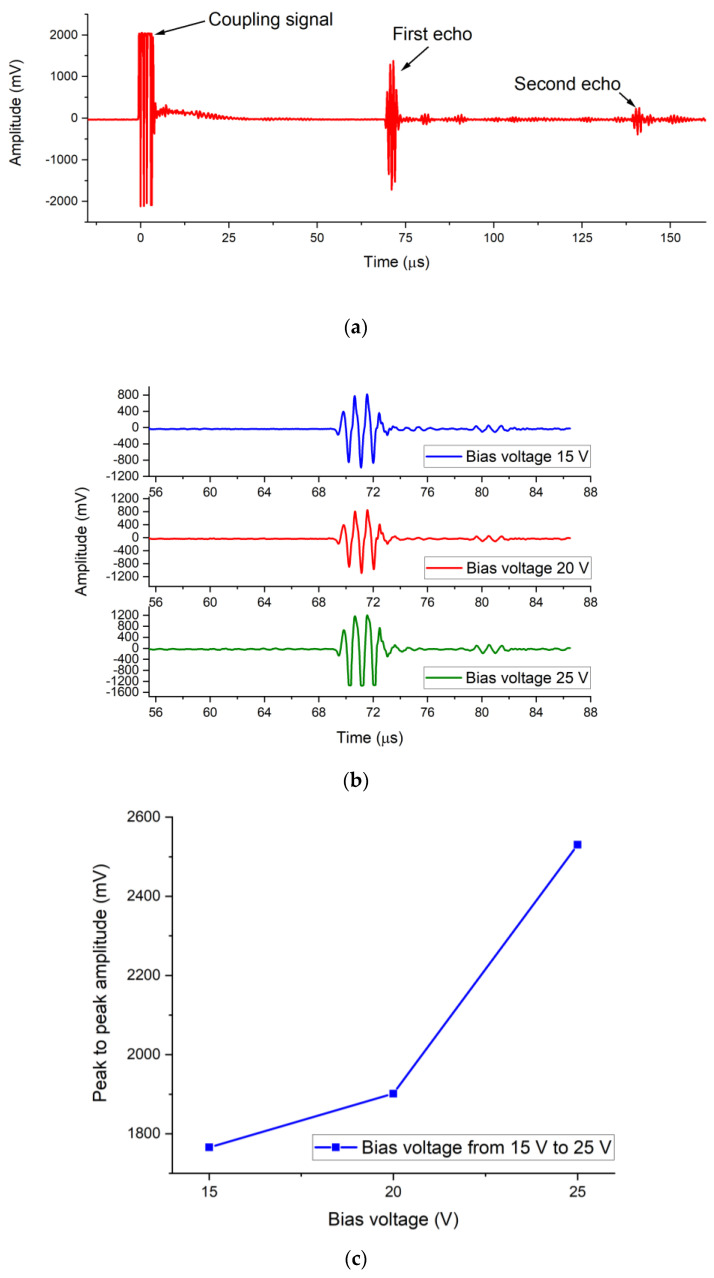
(**a**) Pulse-echo impulse response; (**b**) the first pulse-echo impulse responses when the CMUT device was biased at 15 V, 20 V, and 25 V; (**c**) peak-to-peak amplitude of the first echo signals when the CMUT was driven by different DC bias voltages.

**Table 1 micromachines-12-00516-t001:** Capacitive micromachined ultrasound transducer (CMUT) device parameters.

Parameter	Value
Device size	8800 μm × 8800 μm
Number of cells per device	900 (30 × 30)
Cell size	200 μm × 200 μm
Vacuum cavity diameter	180 μm
Vacuum cavity gap distance	0.65 μm
Oxide silicon Insulating dielectric thickness	0.15 μm
Silicon membrane thickness	2.83 μm
Silicon substrate thickness	400 μm

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
