# Peer review of "Wafer-Bonding Fabricated CMUT Device with Parylene Coating"

_micromachines, 2021, doi:10.3390/mi12050516_

Round 1

Reviewer 1 Report

This is a well-written paper in an area of current interest, with a reasonably good use of the literature. I recommend publication after the following comments are addressed:

  1. Overall, I think the study has merit, but I don’t think the title is fully reflective of the content. There may be potential for underwater imaging after further development, but the content of the article does not fully justify the title it has been given. No explicit imaging results are presented, so I’d advise adapting the title to “Wafer-Bonding Fabricated CMUT Device with Parylene Coating”, also reviewing the content with respect to the imaging. It is fine to identify the potential of the technology, but the title cannot include the term “underwater imaging”, if there are no such results included in the article.
  2. I think more information regarding the design process should be included – I assume the specifications are set as they are to ensure the required dynamic characteristics, but this should be demonstrated.
  3. Can the biasing voltages be considered a significant restriction? I assume there can be issues in the receive mode, and nonlinear drive phenomena?
  4. “between two silicon surfaces at high temperatures” (Line 96) Were these temperatures measured, and are there any additional risks to the electronics from this? I notice a citation is included here too.
  5. I’m not clear on why (specifically) Parylene-C was chosen. A brief justification of this in Section 3 would suffice.
  6. A little explanation of the phenomenon around 10V biasing voltage in Figure 5 would be useful. It appears to be consistent across both conditions.
  7. In the characterisation shown in Section 3.2, was the radiation pattern of the ultrasound measured? Minor physical inconsistencies (which can be unavoidable), can make the optimal measurement axis off-centre.
  8. “Presently, the frequently used center frequency of CMUTs is commonly 10 MHz. Such frequency is challenging to obtain a high-resolution image.” (Wang et al., Capacitive micromachined ultrasound transducers for intravascular ultrasound imaging, Microsystems & Nanoengineering 6, no. 1 (2020): 1-13) How does this observation, published very recently, fit with the device specifications and realistic potential for practical application in ultrasound imaging, including frequency of 1.1 MHz, proposed here?
  9. In Section 3.3, how well matched were the CMUTs in terms of the dynamics? Also, I would recommend providing the distance of separation, if only to ensure others could properly replicate the study.
  10. “basically the same” (Line 201) I think this should be reworded – either they are the same or they are not.
  11. Specify the distance of separation in Figure 10(a), and I think there needs to be a little more examination of why the amplitude-bias trend looks as it does for Figure 11(c).

Some additional minor comments are included below for the authors to consider:

  1. “rarely considered underwater ultrasound imaging applications” (Line 47) Whilst I agree that imaging in this context has mostly been applied for medical procedures, I can still find many sources investigating this with CMUTs in the past few years.
  2. “PECVD or LPCVD” (Line 52) Although they might be obvious, the acronyms should still be explained here (and for other similar instances in the article, such as “SOI” in Line 101 and “CMP” in Line 111).
  3. “The size of a device for underwater imaging is much larger than that of the medical CMUT” (Lines 53-54) This is true, but then CMUTs are often configured in arrays.
  4. “due to stress-related problems” (Lines 57-58) I agree, but I think it would help to identify what some of these are.
  5. Although the supporting text is well written, I’d include a few annotations/labels into Figure 3 to clearly differentiate the layers and their associated materials.
  6. “Voltage” is misspelled in the abscissa of Figure 5.
  7. I’d consider moving the labels on Figure 6(b) outside the figure itself, as they were a little difficult to read. This also applies to Figures 8(b) and 10(b).

Reviewer 2 Report

The presented paper considers the fabrication and characterization of a CMUT device with a 2.83-µm thick silicon membrane dedicated for underwater imaging applications. This application is important since for a 3D underwater imaging sonar one needs to develop efficient 2D transducer arrays. CMUTs are the best candidates for this. The majority of the CMUT presented in the literature deals with medical applications and operates at rather high frequencies above 3 MHz. The main issue in applying CMUT for underwater applications is the necessity to decrease the operating frequency which involves augmentation of the CMUT membrane thickness. The standard silicon IC fabrication technology suffers from stress-related issues while fabricating thick membranes.

The authors present here the CMUT device that was fabricated using silicon fusion wafer-bonding technology by a direct bond between two silicon surfaces at high temperatures. This showed features of high bond strength and extreme hermeticity. The principle of the silicon fusion wafer-bonding was presented in detail in the presented works by the authors. Here, the fabrication and the design of the CMUT are very clearly presented. After, characterization results for CMUT capability for transmission of an ultrasound wave, receiving of an ultrasound wave, and a pulse-echo measurement capability are presented. The experimental results are viable and explicitly explained.

In the overall review’s opinion, this is a high-quality research paper written in perfect English. The CMUT underwater application presented here is of significant importance. Thus, the paper could be accepted for publication in the present form.

However, several minor propositions could be done, the authors are free to accept them or not.

Line 17. “depositing processes using IC processes”… Possibly, it would be better to say “depositing processes using standard IC fabrication technology” (avoiding the “processes “ word repetition)

Line 47. “and integrated circuit (IC) integration” … Possibly, it would be better to say “and integrated circuit (IC) incorporation” (as above, avoiding the “integration“ word repetition)

Spelling issue in Fig. 5 caption: “Bias Vol(a)tage”.

Line 167. “Sinusoidal impulses consisted of five sine waves” … It’s better to say “Sinusoidal impulses consisted of five sine cycles”.

Round 2

Reviewer 1 Report

I thank the authors for adapting the draft and considering my comments. I think a few points need to be revisited before publication, which I think would help the final quality of your article that is clearly based on good research. Please use your ‘Response to Reviewer 1 Comments’ document for reference:

  • Point 2: I understand your point about the focus of the paper, but the abstract states that a CMUT device is proposed and fabricated. I think it is reasonable to expect the design process to be outlined, even as a bullet point list. The dynamic characteristics must be tailored in some way, and I think that is important for other researchers to know how to replicate the study.
  • Point 6: I appreciate you might not know the reason, but I think some speculation of this would be very useful for other researchers.
  • Point 7: I think you should incorporate your response to this point into your article.
  • Point 9: Your response doesn’t answer my question regarding how well matched the CMUTs are. This is important for other researchers hoping to replicate the study.
  • Point 11: You include the distance as I recommended, but you have not offered any insight of why the amplitude-bias trend looks as it does in Figure 11(c). (Somewhat related to the above, the discussion of some key results was missing in your original draft and I think this is another example).
  • Point 15: You state in your response that you agree, but I can’t see what these might be in your revision.
